# UNIVERSAL LEARNING APPROACH FOR ADVERSARIAL DEFENSE

## ABSTRACT

Adversarial attacks were shown to be very effective in degrading the performance of neural networks. By slightly modifying the input, an almost identical input is misclassified by the network. To address this problem, we adopt the universal learning framework. In particular, we follow the recently suggested Predictive Normalized Maximum Likelihood (pNML) scheme for universal learning, whose goal is to optimally compete with a reference learner that knows the true label of the test sample but is restricted to use a learner from a given hypothesis class. In our case, the reference learner is using his knowledge on the true test label to perform minor refinements to the adversarial input. This reference learner achieves perfect results on any adversarial input. The proposed strategy is designed to be as close as possible to the reference learner in the worst-case scenario. Specifically, the defense essentially refines the test data according to the different hypotheses, where each hypothesis assumes a different label for the sample. Then by comparing the resulting hypotheses probabilities, we predict the label and detect whether the sample is adversarial or natural. Combining our method with adversarial training we create a robust scheme which can handle adversarial input along with detection of the attack. The resulting scheme is demonstrated empirically.

## 1 INTRODUCTION

Deep neural networks (DNNs) have shown to have state-of-the-art performance in many machine learning tasks (Goodfellow et al., 2016a). Despite the impressive performance, it has been found that DNNs are susceptible to adversarial attacks (Szegedy et al., 2013; Biggio et al., 2013). These attacks cause the network to under-perform by adding specially crafted noise to the input, such that the original and modified inputs are almost indistinguishable.

In the case of an image classification task, adversarial attacks can be characterized according to three properties (Carlini et al., 2019): the adversary goal, the adversary capabilities, and the adversary knowledge. (i) The adversary goal may be to simply cause misclassification. This is referred to as an untargeted attack. Alternatively, the goal can be to have the model misclassify certain examples from a source class into a target class of its choice. This is referred to as a targeted attack. (ii) The adversary capabilities relate to the strength of the perturbation the adversary is allowed to cause the data, i.e., the distance between the original sample and the adversarial sample under a certain distance metric must be smaller than $\epsilon$. (iii) The adversary knowledge represents what knowledge the adversary has about the model. This can range from a fully white-box scenario, where the adversary has complete knowledge about the model and its parameters, to a black-box scenario where the adversary does not know the model and only has a limited number of queries on it.

Out of the many different adversarial attack algorithms, gradient optimization-based attacks are known to be the most powerful kind of attacks (Carlini et al., 2019). In these attacks, the adversary uses the gradients of a loss function with respect to the input in order to find the perturbation that minimizes the performance of the network. Such attacks include, among others, Fast Gradient Sign Method (FGSM) (Goodfellow et al., 2014), Projected Gradient Descent (PGD) (Madry et al., 2017) and CW-attack (Carlini & Wagner, 2017).

Adversarial defenses can be separated into two categories according to their goal: to increase robustness against adversarial examples (Madry et al., 2017) and to detect whether an example is natural

or adversarial (Li & Li, 2017; Feinman et al., 2017). While recent years have shown growing interest in understanding and defending against adversarial examples, no solution has been found yet to white-box settings, where the adversary has full access to the network.

The universal prediction framework considers both the stochastic setting in which the true relation between the features and labels is given by a stochastic function, and the individual setting in which no probabilistic connection between the data and the labels is assumed. The Predictive Normalize Maximum Likelihood (pNML) as the universal learning solution is proposed for the batch learning in the individual settings (Fogel & Feder, 2018a; Bibas et al., 2019a;b). The pNML scheme gives the optimal solution for a min-max game where the goal is to be as close as possible to a reference learner, who knows the true label but is restricted to use a learner from a given hypothesis class. Further elaboration on the pNML learner is presented in section 3.

We propose the *Adversarial pNML* scheme as a new adversarial defense and detector. Adversarial pNML improves DNNs robustness against adversarial attacks and allows the detection of adversarial samples. Based on the pNML, we restrict the genie learner, a learner who knows the true test label, to perform minor refinements to the adversarial examples. Our method essentially uses an existing model trained with adversarial training. Then, we compose a hypothesis class. Each member in the class assumes a different label for the input. The member refines the input sample based on the assumed label. Finally, by comparing the resulting hypotheses probabilities, we can predict the true label along with detection that the sample is adversarial.

In this paper, we propose a novel scheme that uses a simple targeted adversarial attack as a method of refinement. By performing that simple refinement we create a hypothesis class and use it to predict the test label. We demonstrate our method robustness to state-of-the-art adversarial attacks as PGD (Madry et al., 2017) for MNIST and CIFAR10 datasets. We show that by combining adversarial sample detection, which is an inherent property of our method, we manage to overcome a defense-aware adaptive attack. Unlike existing methods that attempt to remove the adversary perturbation (Samangouei et al., 2018; Song et al., 2017; Guo et al., 2017), our method is unique in the sense it does not remove the perturbation but rather exploits the adversarial subspace properties.

## 2 RELATED WORK

In this section, we mention related works on common attack methods, various type of defence and the properties of the adversarial subspace.

**Attack Methods** One of the simplest attacks is FGSM that was introduced by Goodfellow et al. (2014). Let $w_0$ be the parameters of the trained model, $x$ be the test data and $y$ its corresponding label, $L$ the loss function of the model, $x_{adv}$ the adversary input and $\epsilon$ specifies the maximum $l_\infty$ distortion $||x - x_{adv}||_\infty \leq \epsilon$. First, the sign of the loss function gradients are computed with respect to the image pixels, then after multiplying the signs by $\epsilon$, it is added to the original image to create an adversary untargeted attack:

$$x_{adv} = x + \epsilon \cdot sign \nabla_x L(w_0, x, y_{true}). \tag{1}$$

It is also possible to improve classification chance for a certain label $y_{target}$ (targeted attack):

$$x_{adv} = x - \epsilon \cdot sign \nabla_x L(w_0, x, y_{target}). \tag{2}$$

A multi-step variant of FGSM that was used by Madry et al. (2017) is called Projected Gradient Descent (PGD). Denote $\alpha$ as the size of the update, for each iteration, an FGSM step is executed

$$x_{adv}^{t+1} = x_{adv}^t + \alpha \cdot sign(\nabla_x L(w_0, x, y_{true})), \quad 0 \leq t \leq T. \tag{3}$$

The number of iterations is predetermined. For each sample, PGD attack creates multiple different adversarial samples by randomly choosing multiple different starting points $x_{adv}^0 = x + u$ where $u \sim U[-\epsilon, \epsilon]$. Then the sample with the highest loss is chosen. The strength of the attack is defined by the allowed distance between final adversarial input and the original image $||x - x_{adv}^T|| \leq \epsilon$.

**Defence Methods** Several defense methods have been suggested to increase DNNs robustness. The most prominent defense is adversarial training, which augments the trainset to include adversarial examples (Goodfellow et al., 2014). Other methods include a transformation of the input as

suggested in Guo et al. (2017) where the input is reconstructed from its lower resolution version. A different approach aims only to detect adversarial examples. Such a method is suggested by Feinman et al. (2017), where detection is performed by measuring the confidence for a given input using bayesian uncertainty estimation, available in dropout neural networks.

**Adversarial Subspace** Szegedy et al. (2013) states that adversarial examples represent low-probability pockets in the manifold which are hard to find by randomly sampling around the given sample. Later works show that while adversarial subspace is low-probability they inhabit large and contiguous regions (Goodfellow et al., 2014; Madry et al., 2017). Tabacof & Valle (2016) showed that the adversarial subspaces are less stable compared to the true data subspaces. Our scheme takes advantage of these characteristics in order to improve the robustness of the model.

## 3 PRELIMINARIES: THE PREDICTIVE NORMALIZE MAXIMUM LIKELIHOOD

We now describe the universal learning problem and its associated solution: the predictive Normalized Maximum Likelihood (pNML). In the case of supervised machine learning, a trainset consisting of $N$ pairs of examples, $z^N = \{(x_i, y_i)\}_{i=1}^N$, is given to a learner, where $x \in \mathcal{X}$ is the data and $y \in \mathcal{Y}$ is the label. The goal of the learner is to predict the label $y$ given a new test data $x$. More formally, the learner attempts to minimize a loss function that measures the accuracy of the prediction. In the information-theoretic framework, a prediction is done by assigning a probability distribution $q(\cdot|x)$ to the unknown label. The performance of the predictor is evaluated using the log-loss function

$$L(q; x, y) = -\log q(y|x). \tag{4}$$

For the problem to be well-posed we must make further assumptions on the class of possible models or "hypothesis" that is used in order to find the relation between x and y. Denote $\Theta$ as a general index set, this class is a set of conditional probability distributions $P_\Theta = \{p_\theta(y|x), \ \theta \in \Theta\}$.

Another assumption, required to solve the problem, is on how the data and the labels are generated. The most common setting in learning theory is Probably Approximately Correct (PAC), established in (Valiant, 1984) where $x$ and $y$ are assumed to be generated by some source $P(x, y) = P(x)P(y|x)$. $P(y|x)$ is not necessarily a member of $P_\Theta$. Another possible setting for the learning problem, recently suggested by Fogel & Feder (2018a) and Merhav & Feder (1998), is the individual setting, where the data and labels of both the training and test are specific and individual values. In this setting, the goal is to compete with a "genie" or a reference learner that knows the desired label value but is restricted to use a model from the given hypotheses class $P_\Theta$, and that does not know which of the samples is the test. This learner then chooses:

$$\hat{\theta}(z^N, x, y) = \arg\max_\theta \left[ p_\theta(y|x) \cdot \Pi_{i=1}^N p_\theta(y_i|x_i) \right]. \tag{5}$$

The log-loss difference between a universal learner $q$ and the reference is the regret:

$$R(q; z^N, x, y) = \log \frac{p_{\hat{\theta}(z^N, x, y)}(y|x)}{q(y|z^N, x)}. \tag{6}$$

As advocated in Fogel & Feder (2018b), the chosen universal learner solves:

$$\Gamma = R^*(z^N, x) = \min_q \max_y R(q; z^N, x, y). \tag{7}$$

This min-max optimal solution, termed Predictive Normalized Maximum Likelihood (pNML), is obtained using "equalizer" reasoning, following (Shtar'kov, 1987):

$$q_{\text{pNML}}(y|x; z^N) = \frac{\max_{\theta \in \Theta} p_\theta(y|z^N, x, y)}{\sum_{y \in \mathcal{Y}} \max_{\theta \in \Theta} p_\theta(y|z^N, x, y)}. \tag{8}$$

Its corresponding regret, independent of the true $y$, is:

$$\Gamma = \log \sum_{y \in \mathcal{Y}} \max_{\theta \in \Theta} p_\theta(y|z^N, x, y). \tag{9}$$

The pNML has been derived for several model classes in related works such as 1-D perceptron in Fogel & Feder (2018b) and linear regression in Bibas et al. (2019a). They show that the regret can serve as a pointwise measure of learnability for the specific training and data sample. Bibas et al. (2019b) also reported that the pNML regret can be used to detect out of distribution test samples.

## 4 ADVERSARIAL PNML

Recall the pNML solution in equation 8. Intuitively, for each of the possible labels $y \in Y$ the pNML chooses a learner from a hypothesis class $\Theta$ that maximizes the probability of that label. The probability of the respective label is then normalized by the normalization factor, which is the same for all labels. Denote the unnormalized probability of the true label as the "genie"

$$\max_{\theta \in \Theta} p_\theta(y|z^N, x, y_{true}). \tag{10}$$

The "genie" is the best learner one can attain when knowing the test label. The main issue is how to select the hypothesis class. A "good" hypothesis class should be large enough so that the genie could achieve good performance while avoiding overfitting of the other class members such that the overall regret (the distance from the genie learner as mentioned in equation 9) stays small.

We propose a novel hypothesis class by adding refinement stage before a pretrained DNN model $w_0$. The refinement stage changes the test sample with respect to the model $w_0$ and a certain label $y$

$$x_{refine}(x, y) = x - \lambda \cdot sign(\nabla_x L(w_0, x, y)) \tag{11}$$

where $\lambda$ is the refinement strength. The hypothesis class that we consider is therefore

$$\Theta = \left\{ p_{w_0}(y|x_{refine}(x, y_i)), \quad \forall y_i \in Y \right\}. \tag{12}$$

Each member in the hypothesis class produces a probability assignment by adding a perturbation that strengthen one of the possible value of the test label.

Our Adversarial pNML scheme consists the following steps: At first, we train a DNN model $w_0$. The training could be adversarial-training (Goodfellow et al., 2014) or normal training. Then, we produce the hypothesis class: Given a test data $x$, we refine it using the trained DNN $w_0$ and one of the possible value of the test label $y_i \in Y$ as in equation 11. We produce a probability assignment for the selected label $y_i$ by feeding the refined test data $x_{refine}$, to the trained DNN $w_0$

$$p_i = p_{w_0}(y_i|x_{refine}(x, y_i)). \tag{13}$$

The process is repeated for every possible value of the test label. In the end of the process we get a set of predictions (probability distributions), we normalize them and return the Adversarial pNML probability assignment

$$q_{pNML}(y_i) = \frac{1}{C} p_i, \quad C = \sum_{i=1}^{|Y|} p_i. \tag{14}$$

The corresponding regret is the log normalization factor:

$$\Gamma = \log C = \log \sum_{i=1}^{|Y|} p_i. \tag{15}$$

Previous works demonstrated that it is possible to detect adversarial examples by measuring the confidence of the predictions (Feinman et al., 2017). As mentioned in section 3, the regret is the distance from the genie learner, who knows the true label of the test sample. Having high regret means being far from the genie and therefore having low confidence in the prediction. Such a situation is used to detect adversarial examples and is demonstrated in section 6.

We now discuss the reasons behind the choice of the suggested hypothesis class. Let $x$ be a strong adversarial example with respect to the label $y_{target}$, i.e., the DNN model has a high probability mistakenly classifying $x$ as $y_{target}$. We now differentiate between three kind of members in our suggested hypothesis class: refinement towards the true label $y_{true}$, refinement towards the adversary target $y_{target}$ and refinement towards other label $y \notin \{y_{true}, y_{target}\}$.

**Refinement towards the true label**    When perturbing towards the true label $y_{true}$, the refinement moves the adversarial example outside the loss function's local maximum, thus increasing the probability of predicting the true label.

Table 1: Accuracy rate for various adversary attacks and adversary trained models.

| Model \ Attack | MNIST | | | CIFAR10 | | |
|---|---|---|---|---|---|---|
| | None | FGSM | PGD | None | FGSM | PGD |
| Natural | 99.3% | 0.6% | 0.0% | 93.6% | 6.1% | 0% |
| Natural + Ours | 99.2% | 1.6% | 0.0% | 87.2% | 5.9% | 1.0% |
| FGSM | 97.1% | 97.7% | 0.3% | 83.5% | 49.4% | 36.2% |
| FGSM + Ours | 97.2% | 91.4% | 23.3% | 79.4% | 50.1% | 48.8% |
| PGD | 98.3% | 95.8% | 90.5% | 84.3% | 59.5% | 37.4% |
| PGD + Ours | 98.3% | 93.9% | 95.2% | 79.1% | 60.8% | 60.1% |

**Refinement towards the adversary target**    Refinement towards $y_{target}$ is essentially taking another step towards the local maximum of the loss. In the case of a strong adversary, the loss was already converged to the local maximum and another step towards $y_{target}$ would not dramatically increase the probability of $y_{target}$. The loss remains roughly the same and so the probability assignment of the hypothesis. Since the adversarial subspace is relatively small and unstable (see section 2), a weak refinement towards $y_{target}$ could even move the adversarial example outside the local maximum of loss function which effectively reduces the strength of the attack.

**Refinement towards other label**    This refinement effectively applies a weak targeted attack towards a specific label. Therefore, it increases the probability to misclassify the test data as $y$. As mentioned in Section 2, the adversarial examples represent a low-probability subspaces in the manifold which are hard to find. A weak step towards $y$ is unlikely to find a strong adversarial points.

Since the low-probability and instability of the adversarial subspace is crucial for Adversarial pNML, we combine our method with adversarial training, which is known to decrease the size of the adversarial subspace (Madry et al., 2017).

## 5    EXPERIMENTS

In this section we present experiments that test our proposed Adversarial pNML scheme as a defence for adversarial attack. We test the performance on MNIST (LeCun et al., 2010) and CIFAR10 (Krizhevsky et al., 2014) datasets for PGD attack which represents attack that that efficiently maximizes the loss of an example (Madry et al., 2017), and FGSM attack that illustrates a weaker attack.

### 5.1    MNIST DATASET

We follow the model architecture as described in Madry et al. (2017). We use a network that consists of two convolutional layers with 32 and 64 filters respectively, each followed by $2 \times 2$ max-pooling, and a fully connected layer of size 1024. We use three different trainsets that are used in training the baseline approaches: (i) Natural trainset; (ii) Adversarial trainset that was generated by executing FGSM attack on the natural trainset with $\epsilon$ value of 0.3; (iii) Adversarial trainset that was produced by PGD based attack on the natural trainset with 40 steps of size 0.01 with a maximal $\epsilon$ value of 0.3. We train three different models using these three trainsets. The training consists of 100 epochs using stochastic gradient descent (SGD) with a learning rate of 0.01, momentum value 0.9 and weight decay of 0.0001.

We consider the following threat: a white-box $l_\infty$ that is unaware of the defense, with attack strength $\epsilon$ that equals to 0.3. We test the accuracy in the following test cases: natural samples, FGSM attack, and PGD attack. For the PGD attack, the attacker performs 40 steps of size 0.01 with 20 restarts.

Table 1 shows the efficiency of our scheme for the various attacks. First, in case of natural images, when no perturbation is added to the original MNIST dataset, our method obtains approximately the same accuracy as the baselines. In the face of PGD attack, which is the strongest adversary, a model that was trained using FGSM perturbation obtains a significant improvement of 20% in the accuracy rate when using our method, and improvement of 5% on model that was trained with PGD

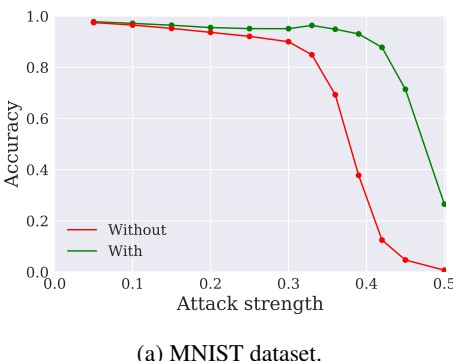

(a) MNIST dataset.

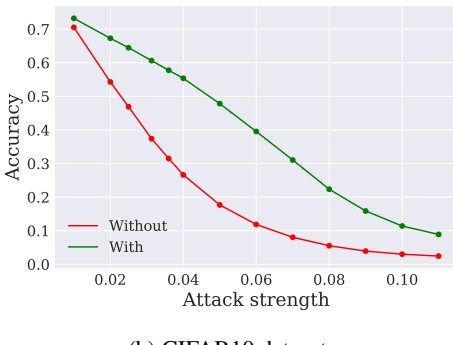

(b) CIFAR10 dataset.

Figure 1: Comparison between a model without our scheme (Without) to the same model with Adversarial pNML scheme (With) for various attack strength $\epsilon$. For each dataset the corresponding model was trained with PGD trainset. (a) MNIST dataset. (b) CIFAR10 dataset.

trainset. On the other hand, in the case of FGSM attack, we see mixed results. When using the FGSM train method we see a reduction of 6% in the accuracy rate and a decrease of 1% in the case of PGD based training. As explained in section 4, our scheme works best when the adversary finds a local maximum of the error function, this scenario is not necessarily the case when using FGSM as it is regarded as a weak attack. Nevertheless, increasing model robustness is first and foremost increasing the robustness in the face of the worst-case attack (Madry et al., 2017). We point out to the fact that for a model trained with PGD trainset, the worst-case attack for our scheme is created by FGSM attack and not PGD attack. This indicates that PGD can fail to find the optimal attack for our scheme. In section 6, we present an adaptive adversary that is explicitly designed to attack our proposed method. We show that we indeed maintain better performance in the worst-case attack.

In order to perform a broader evaluation of the adversarial robustness of our scheme, we investigate the robustness to $l_\infty$ bounded attacks for different attack strengths. We use the PGD based trained model and keep during the training the $\epsilon$ value constant with a value of 0.3. We compare between this model with and without our scheme. For the attack, we use PGD with 50 iterations of size 0.01 with 20 random restarts. We chose the $\lambda$ value to be equal 0.1 for the refinement strength. The results for MNIST dataset appear in figure 1a. Our scheme show significant improvement compared to the same model without Adversarial pNML for all $\epsilon$ values. There is a rapid decrease for $\epsilon$ value of 0.3 for the basic model. The decrease can be explained by attacking with $\epsilon$ that is greater than the one it was trained with. Using our scheme, we able to maintain the performance up to $\epsilon$ value that equals to 0.39.

## 5.2 CIFAR10 DATASET

For CIFAR10 we use a modified version of Wide-ResNet (Zagoruyko & Komodakis, 2016) with each layer being wider by a factor of 10. This DNN contains three residual units with 160, 320, and 640 filters respectively, with a total of 32 layers. As in MNIST dataset, we use three different trainsets to generate three different models: (i) Natural trainset; (ii) Adversarial trainset that was generated by FGSM attack on the natural trainset with $\epsilon$ value that equals to $8/255$; (iii) Adversarial trainset that was produced by using PGD attack on the natural trainset with 20 steps of size $2/255$ with a total $\epsilon$ of $8/255$. We train over 200 epochs using SGD optimizer with a learning rate of 0.001, reducing it to 0.0001 and 0.00001 after 100 and 150 epochs respectively. We also use momentum value of 0.9 and weight decay of 0.0002. We evaluate the performance of our method against the performance of the same model without our scheme for all three trainsets.

We consider the following threat model: a white-box $l_\infty$ that is unaware of the defense with an attack strength of $8/255$. We test the accuracy in the following test cases: natural samples, FGSM attack, and PGD attack. For the PGD attack, we perform 20 steps of size $2/255$ with 1 restart.

We illustrate our results in Table 1. Similar to the MNIST results, we see that our scheme improves the robustness against PGD attack by 23.7% for a model that was trained with PGD trainset. Overall, the robustness against the worst-case attack is improved for all models. On the other hand, the

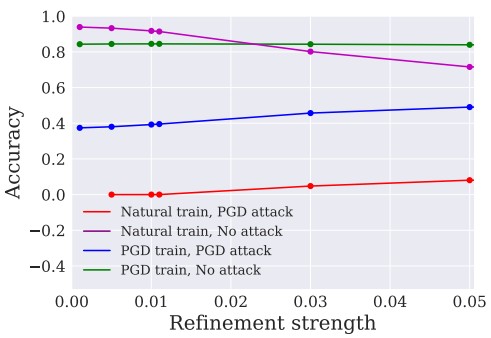

(a) Accuracy rate for different attack strength $\epsilon$

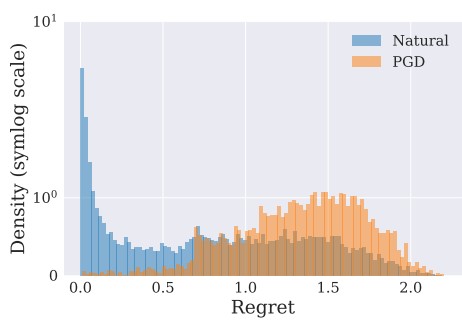

(b) CIFAR10 regret histogram

Figure 2: (a) CIFAR10 performance for different attack strength $\epsilon$. Two models are tested, the first trained with adversarial PGD trainset and the second with natural trainset. We compare the performance of those models with and without our proposed scheme. (b) Regret histogram for correctly classified natural samples and PGD adversarial samples which causes misclassification.

accuracy of natural samples is decreased. This is especially true when the model is trained with natural trainset when our adversarial pNML is used the accuracy drops by $6.4\%$.

In figure 1b, we further explore PGD attacks accuracy as a function of the attack strength $\epsilon$. We use the previous model that was trained with PGD adversarial trainset, during training the $\epsilon$ remains constant with a value of $8/255$. We compare the model robustness against the same model that was integrated with our scheme. Our scheme shows significant improvement, especially around $\epsilon$ that equals 0.05 where we gain almost $30\%$ improvement.

Choosing the refinement strength represents a trade-off between increasing the robustness against adversarial attacks and decreasing the accuracy of natural samples. This trade-off is explored in figure 2a. The advantage of our scheme is displayed when it is combined with adversarial training (training with a PGD attacked trainset). As the refinement strength increases so is the robustness to PGD attack with almost no loss in accuracy for the natural data. On the other hand, when our method was applied to a model trained without adversarial training we see a decrease in accuracy for natural data. As elaborated in section 4 the refinement process is, in fact, a weak targeted adversarial attack toward different labels. Therefore, when a natural sample is inserted to our scheme it becomes weakly adversarial. A model that was trained without adversarial training is susceptible even to weak attacks, hence the performance for natural samples drops. To limit the performance degradation for a non-adversarial trained model we chose a small $\lambda$ value of 0.02, while for the adversarially trained model we chose $\lambda$ value of 0.11.

We demonstrate the ability of the regret to differentiate between correctly classified natural samples and adversarial samples that causes misclassification in figure 2b which shows a clear separation between natural images and adversarial ones.

## 6 ADAPTIVE ADVERSARY

Part of the defense evaluation is to create and test against adaptive adversaries that are aware of the defense (Athalye et al., 2018; Carlini et al., 2019). We design an adaptive adversary for our scheme: We create an end-to-end model that calculates all possible hypotheses in the same computational graph. Then by using gradient-based optimization on the new end-to-end model, we create our adaptive adversarial attack.

The first obstacle with using gradient-based optimization for that end-to-end model is that $sign(\cdot)$ operator that is used in the refinement stage, which sets the gradient to zero during the backpropagation. Athalye et al. (2018) suggested using Backward Pass Differentiable Approximation (BPDA) technique to overcome that problem. In BPDA, we perform forward-pass as usual, but on the backward pass, we replace the non-differentiable part with the identity operator.

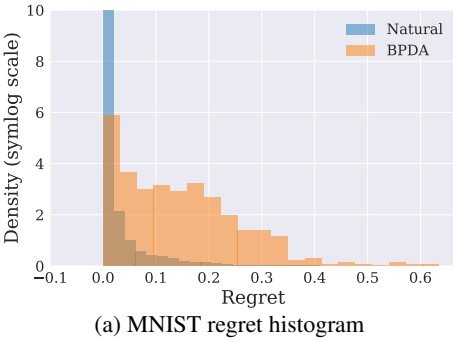
(a) MNIST regret histogram

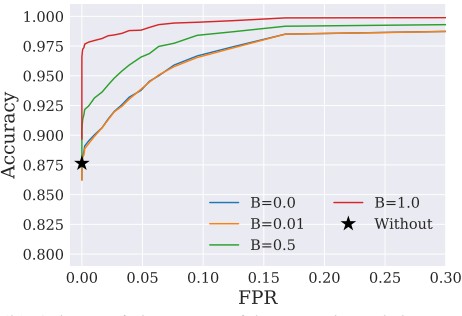
(b) Adversarial pNML with regret based detector

Figure 3: (a) Regret histogram for correctly classified natural samples and BPDA adversarial samples which cause misclassification. In (b) the accuracy represents the ratio between adversarial samples correctly classified or detected and the total number of sample for various $\beta$. The black star point represents the worst-case attack (PGD) against the same model without our defense.

In order to overcome the BPDA attack, we use the inhered detection property of our scheme and we define the following goal (as in Meng & Chen (2017)): Correct classification is made when the true label is assigned to the sample or when an adversarial sample is detected. This new goal gives rise to a new scheme which incorporates adversarial detector in addition to adversarial pNML. We utilize the regret from equation 15 to form an adversarial detector. Figure 3a shows the regret histogram for MNIST dataset. We note that adversarial samples generated by BPDA attack have higher regret values compared to natural samples. Using different regret thresholds, one can control the trade-off between the accuracy of the adversarial testset and the False Positive Ratio (FPR) of the natural testset, i.e., the number of natural samples that are detected as adversarial samples.

Due to the assumption that the attacker is aware of our defense, we propose a second new attack that designed to target our detector and name it *Adaptive Attack*. The Adaptive Attack is designed to minimize the regret to avoid detection. This is done by adding a regularization term to the loss:

$$L(x, y_{true}) = -\log\left(\frac{p_{w_0}(y_{true}|x_{refine}(x, y_{true}))}{\sum_{i=1}^{|Y|} p_i}\right) - \beta \log \sum_{i=1}^{|Y|} p_i. \tag{16}$$

The advantage of our defense is that the attacker have contradictory objectives. On one hand, to avoid detection the attacker needs to minimize the regret $\log\left(\sum_{i=1}^{|Y|} p_i\right)$, and on the other hand to increase the classifier loss $-\log\left(p_{w_0}(y_{true}|x_{refine})/\sum_{i=1}^{|Y|} p_i\right)$ it needs to maximize $\log\left(\sum_{i=1}^{|Y|} p_i\right)$.

We demonstrate the results of the adversarial pNML combined with the regret detector in figure 3b over 3000 MNIST samples. We use the same model as described in section 5.1 trained with PGD adversarial trainset. For the refinement strength we chose $\lambda$ value of 0.03. For the adaptive attack, we examine various $\beta$ values as shown in figure 3b. Our proposed defense is successful against adaptive attack. For regret threshold of $0.19$, we get $91.3\%$ accuracy with FPR of $2\%$ against adaptive attack. This represents a $4\%$ improvement compared to a model without our scheme.

## 7 CONCLUSION AND FUTURE WORK

In this paper we presented the Adversarial pNML scheme for defending DNNs from adversarial attacks by increasing robustness and detection. We showed empirically that our defense increases robustness against unaware white-box attacks for MNIST and CIFAR10 datasets. To overcome adaptive attacks, we successfully use the regret, which is an inherent property of our method, as an adversarial detector in order to enhance our defense.

This work suggests several potential directions for future work. First, we plan to try our scheme against other attacks with different $l_p$-norms. Second, it is interesting to explore other hypothesis classes, one such class is the entire "model class" where instead of refining the sample we refine the model according to different hypotheses.

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
