# OpenReview forum: "Universal Learning Approach for Adversarial Defense"
_ICLR.cc/2020/Conference — Reject_

### Official Review · AnonReviewer1 · 2019-10-22
**Official Blind Review #1**

**Rating:** 3

**Review:**

In this paper, the authors proposed the Adversarial predictive normalize maximum likelihood (pNML) scheme to achieve adversarial defense and detection.
The proposed method is an application of universal learning, which is compatible with existing adversarial training strategies like FGSM and PGD.
However, the experimental results indicate that the proposed method is more suitable for the models trained under PGD-based attacks.
According to the analysis shown in the paper, the proposed method works best when the adversary finds a local maximum of the error function, which makes it more robust to strong attacks.
It seems that the proposed work is a good attempt that applies universal learning to adversarial training, but more experiments are required to support its usefulness and effectiveness, especially for the weak attack like FGSM. Additionally, I would like to see more discussions about the limitations of the proposed method.

Minors:
In Figure 2, I would like to see the results related to FGSM.

**Experience Assessment:**

I do not know much about this area.

**Review Assessment: Checking Correctness Of Derivations And Theory:**

I assessed the sensibility of the derivations and theory.

**Review Assessment: Checking Correctness Of Experiments:**

I assessed the sensibility of the experiments.

**Review Assessment: Thoroughness In Paper Reading:**

I read the paper at least twice and used my best judgement in assessing the paper.

---

> ### Author Response · Authors · 2019-11-13
> **Response to reviewer1**
>
> Thank you for your kind and constructive review.
>
> We would like to address your points in the same order:
>
> 1.	“Additionally, I would like to see more discussions about the limitations of the proposed method.”
> Answer: Our method has 4 main drawbacks:
> 1.	As you noted, it is less effective against weak attacks.
> 2.	It causes a drop in natural accuracy.
> 3.	It relies on having a model trained with adversarial examples. This prevents us from testing our method on imagenet since it requires a lot of computational power for training that we do not possess at the moment.
> 4.	Inference time is increased - instead of doing a simple forward-pass our method requires to do a forward-backward pass for the refinement and then another forward pass.
> We counter the first 2 drawbacks by incorporating adversarial training which gives robustness against weak adversarial attacks and limits the natural accuracy drop.
>
> 2.	“In Figure 2, I would like to see the results related to FGSM.”
> Answer: In https://imgur.com/a/neoLND2 you would find figures corresponding to figure 2a but with a model trained with FGSM adversarial examples and tested against Natural, FGSM and PGD samples. Regarding figure 2b, take a look at https://imgur.com/a/ANKWxko - it shows FGSM regret histogram in addition to PGD and BPDA for MNIST dataset.

---

### Official Review · AnonReviewer2 · 2019-10-23
**Official Blind Review #2**

**Rating:** 1

**Review:**

Summary:
This paper focuses on the area of adversarial defense -- both to improve robustness and to detect adversarially perturbed images. They approach the problem from the universal prediction / universal learning framework.
Modivation:
I had a hard time understanding the motivation of this work -- specifically the connection to the universal learning framework which to be fair I am unfamiliar with. In the absence of the universal learning framework formalism the method proposed here is quite simple and in my opinion clever -- create a new prediction based on performing an adversarial attack to each target class. What value does universal learning bring to this?
Second, I do not follow the intuition for the chosen hypothesis class -- why work off of refined images in the first place? Is there some reason to believe this will improve robustness?
Finally, the view that adversarial defense and attack are important topics to explore is under some debate. I am not considering this as part of my review but I would encourage the authors to look at [1].
Writing:
The writing was clear and typo free.
Experiments:
Overall the experiments seemed inconclusive.
Section 5 shows robustness against the unmodified / unrefined model (the attacks are done on the base model not the refined model). Given that these attacks are performed against the unmodified model then evaluated on the modified model the results seem a bit unfair / harder to interpret. The authors note this, and in Section 6 explore the "Adaptive Adversary" setting.
The results presented are performed on Mnist and Cifar10. Overall the results were not convincing to me. Table 1 shows mixed performance -- a drop in natural accuracy in all cases, decreases in FGSM. The main increase in performance is in the PGD. This was noted, but understanding in more depth why this method helps here will hopefully lead to improved performance in FGSM as well.
Figure 2a shows very weak correlations. Figure 2b seems promising but also not necessarily a surprise given that the adversarial examples are generated against the base model and not the refined model.
For section 6, one risk is that the BPDA attack doesn't successfully work. Having some more proof that the attacks presented here are strong would greatly improve the work.
Larger scale experiments would of course be nice and strengthen the paper but more importantly it would be great to see some form of toy example or demonstration of the principle improving robustness as well over just results. Something to probe the mechanism of action for example.
Finally, having some comparisons to other defense strategies would improve this paper.
Rating:
Given the gap between the universal learning framework and the method proposed, as well as the inconclusive experiments at this point I would not recommend the paper for acceptance.

[1] https://arxiv.org/abs/1807.06732

**Experience Assessment:**

I do not know much about this area.

**Review Assessment: Checking Correctness Of Derivations And Theory:**

I assessed the sensibility of the derivations and theory.

**Review Assessment: Checking Correctness Of Experiments:**

I assessed the sensibility of the experiments.

**Review Assessment: Thoroughness In Paper Reading:**

I read the paper at least twice and used my best judgement in assessing the paper.

---

> ### Author Response · Authors · 2019-11-13
> **Response to reviewer2**
>
> Thank you for your kind and constructive review.
>
> We would like to address your points in the same order:
> 1.“I had a hard time understanding the motivation of this work - specifically the connection to the universal learning framework. In the absence of the universal learning framework formalism the method proposed here is quite simple and in my opinion clever - create a new prediction based on performing an adversarial attack to each target class. What value does universal learning bring to this?”
> Answer: You are right in the sense that this isn’t the classic universal learning framework where the hypothesis class is the model itself. Nevertheless, the universal training framework was the main motivation for our work, for example, the fact that the regret can be used as a confidence measure is directly related to previous works described in the end of Section 3.
> 2.“I do not follow the intuition for the chosen hypothesis class - why work off refined images in the first place? Is there some reason to believe this will improve robustness?”
> Answer: When a successful attack occurs, the refinement increases the probability of the true label much more than the adversarial targeted label which is why adversarial pNML works. This happens because the loss of a successful adversarial sample is very likely, already converged to a local-maxima, therefore refinement towards the adversarial target label won’t dramatically increase target probability. For more details please refer to the end of Section 4 and to the first answer given to reviewer 3.
> 3.“Adversarial defense and attack are not very important, look at [1]... ”
> Answer: Thank you for bringing this paper to our attention. We do find it interesting and we will look into it in more detail.
> 4.“Experiments seemed inconclusive - Table 1 shows mixed performance - a drop in natural accuracy in all cases, decreases in FGSM. (Understanding in more depth why this method helps here will hopefully lead to improved performance in FGSM as well.”
> Answer: The reason our method is less successful in defending against FGSM attack is that FGSM is considered a weak adversarial attack. The loss of adversarial samples created by FGSM attack is not converged to the loss local-maxima and therefore the refinement is less helpful (see answer 2).
> While table 1 is focused on adversarial robustness we could incorporate a detection scheme as we did for the adaptive adversary to improve FGSM results aswell.
> 5.“Figure 2a shows very weak correlations.”
> Answer: Figure 2a demonstrates why it is necessary to combine our method with adversarial training. For PGD trained model we see an improvement against PGD  attacks (blue curve) while losing almost no accuracy for natural samples (green curve). The weak correlation to natural samples accuracy is a good thing, it means that we are able to further increase the refinement strength without degrading performance for natural samples.
> 6. Figure 2b seems promising but also not necessarily a surprise given that the adversarial examples are generated against the base model and not the refined model.
> Answer: Generating BPDA attack for CIFAR10 network is computationally extremely complex. However, we can demonstrate that for BPDA attack on MNIST, it is possible to differentiate between normal samples and BPDA samples using the risk - https://imgur.com/a/ANKWxko.
> 7. For section 6, one risk is that the BPDA attack doesn't successfully work. Having some more proof that the attacks presented here are strong would greatly improve the work.
> Answer: In https://imgur.com/a/mUg5LeD you would find a graph that compares BPDA attack to PGD attack with and without our scheme. The comparison was done for MNIST dataset with various attack strengths. For our scheme, BPDA attack is more successful than PGD for all attack strengths, this shows BPDA attack is indeed strong. It is also clear that for most attack strengths, our scheme is more robust than a model without our scheme (red curve).
> While our scheme shows robustness even in the face of BPDA attack we cannot rule out the possibility of an even stronger attack, but this argument is true for almost all the research in the field of adversarial defense (see https://arxiv.org/abs/1902.06705).
> 8.	“…it would be great to see some form of toy example or demonstration of the principle improving robustness as well over just results. Something to probe the mechanism of action for example.”
> Answer: Please refer to the first answer given to Reviewer 3, in it, we demonstrate empirically why adversarial pNML works.
> 9.	Have state-of-the-art defense methods been compared?
> Answer: We consider adversarial training with strong adversarial examples (such as PGD) to be a state-of-the-art defense as presented in Madry et al. paper https://arxiv.org/abs/1706.06083. This claim also appears in the paper you referenced: “The current state-of-the-art defense for the standard rules on the MNIST dataset is due to Madry et al.”

---

### Official Review · AnonReviewer3 · 2019-10-24
**Official Blind Review #3**

**Rating:** 3

**Review:**

This paper proposes an adversarial pNML scheme for adversarial defence and adversarial example detection. The idea is very intuitive and pNML is adopted from the literature work for the purpose of adversarial defence.

The authors provided some explanation on why the adversarial pNML should work. The reasoning is quite intuitive, lacking of thorough justification.  The authors may consider using experiments to provide empirical justifications for the explanations.

The proposed method is heavily dependent on previous works. The section 6 adaptive adversary part is not clear.  How to do the adaptive attack based on Eq.(16)? Maximizing the loss in Eq.(16)?  How to determine the threshold for adversarial example detection?

The experimental results in Table 1 seems to be very good. However, have the state-of-the-art defence methods been compared?


**Experience Assessment:**

I do not know much about this area.

**Review Assessment: Checking Correctness Of Derivations And Theory:**

I did not assess the derivations or theory.

**Review Assessment: Checking Correctness Of Experiments:**

I assessed the sensibility of the experiments.

**Review Assessment: Thoroughness In Paper Reading:**

I made a quick assessment of this paper.

---

> ### Author Response · Authors · 2019-11-13
> **Response to reviewer3**
>
> Thank you for your kind and constructive review.
>
> We would like to address your points in the same order:
> 1.“The authors provided some explanation on why the adversarial pNML should work. The reasoning is quite intuitive, lacking of thorough justification. The authors may consider using experiments to provide empirical justifications for the explanations. “
>
> Answer: We follow your recommendation and add empirical experiments to support the claims made in Section 4 regarding why adversarial pNML should work.
> In the provided link https://imgur.com/a/hcSyx7f you would find histograms showing the probabilities difference before and after refinement. The results are calculated over MNIST for a model trained with PGD adversarial samples and tested with PGD adversarial samples.
> We present 3 histograms, one for refinement towards the true label, the second for refinement towards the adversary target label and the third for refinement towards the other labels (see Section 4 for more details):
> a.	True label – The first histograms present the true label probability difference between the refined sample, $x_{refine}(x,y_{true})$, and original sample $x$. We divide the samples into 2 groups, the first where the true label is also the predicted label (Correct – True label) and the second where the true label isn’t the predicted label (Incorrect – true label). We see that the refinement increases the true label probabilities, especially when the true label isn’t the predicted label.
> b.	Adversary Target label – The second histogram presents the target label (the label the adversary promotes) probability difference between the refined sample, $x_{refine}(x,y_{target})$, and original sample $x$. We divide the samples into 2 groups, the first where the adversary was unsuccessful (the predicted label is the true label) and the second where the adversary was successful (the predicted label is the adversary target label). As explained in Section 4 - Refinement towards the adversary target, in case of a strong adversary (which in our case is successful adversary) the loss is already converged to the local-maxima, therefore refinement towards $y_{target}$ can sometimes decrease the probability as seen in the histogram.
> c.	Other label - The first histograms present the true label probability difference between the refined sample, $x_{refine}(x,y_{other})$, and the original sample $x$. We see that the refinement increases the probabilities, but since labels belonging to that category have a low probability, to begin with, this increase in probability won’t cause misclassification.
> In short, when a successful attack occurs, the refinement increases the probability of the true label much more than the adversarial targeted label which is why adversarial pNML works. This happens because the loss of a successful adversarial sample is already converged to a local-maxima, therefore refinement towards the adversarial target label won’t dramatically increase target probability.
>
> 2.	“The section 6 adaptive adversary part is not clear. How to do the adaptive attack based on Eq.(16)? Maximizing the loss in Eq.(16)?”
> Answer: By Maximizing the loss in Eq (16) using an iterative method such as PGD on the end-to-end model we attempt to maximize the loss to cause misclassification while minimizing the regret to avoid detection. The first term in Eq.(16) is responsible for the loss of the end-to-end model and the second term is a regularization over the regret.
>
> 3.“ How to determine the threshold for adversarial example detection?
> Answer: We determine the threshold of the detection and the trade-off parameter $\beta$ of the adaptive attack by a min-max game - for each threshold value we test against multiple $\beta$ values, the threshold that gives the best accuracy for the worst-case $\beta$ is selected.
>
> 4. “Have state-of-the-art defense methods been compared?”
> Answer: We consider adversarial training with strong adversarial examples (such as PGD) to be a state-of-the-art defense as presented in Madry et al. paper https://arxiv.org/abs/1706.06083. This claim is also repeated in the paper reviewer 2 referenced (https://arxiv.org/abs/1807.06732): “The current state-of-the-art defense for the standard rules on the MNIST dataset is due to Madry et al.”

---

### Decision · Program_Chairs · 2019-12-19

**Decision:**

Reject

**Comment:**

The reviewers attempted to give this paper a fair assessment, but were unanimous in recommending rejection.  The technical quality of motivation was questioned, while the experimental evaluation was not found to be clear or convincing.  Hopefully the feedback provided can help the authors improve their paper.